# MOSPA: Human Motion Generation Driven by Spatial Audio

**Shuyang Xu**[*,1]**, Zhiyang Dou**[*,†,1]**, Mingyi Shi**[1]**, Liang Pan**[1]**, Leo Ho**[1]**, Jingbo Wang**[2]**,
**Yuan Liu**[3]**, Cheng Lin**[4]**, Yuexin Ma**[5]**, Wenping Wang**[†,6]**, Taku Komura**[†,1]

[1]The University of Hong Kong    [2]Shanghai AI Lab
[3]The Hong Kong University of Science and Technology
[4]Macau University of Science and Technology
[5]ShanghaiTech University    [6]Texas A&M University

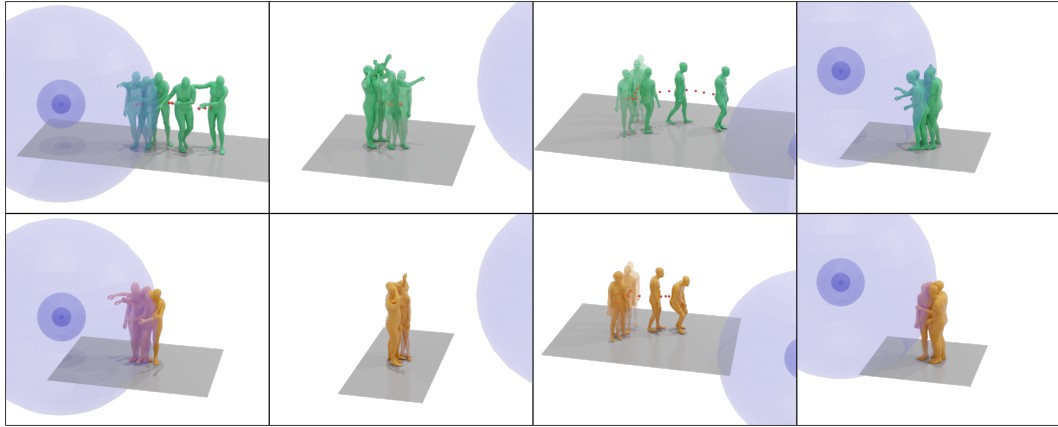

Figure 1: We introduce a novel human motion generation task centered on spatial audio-driven human motion synthesis. Top row: We curate a novel *S*patial *A*udio-Driven Human *M*otion (SAM) dataset, including diverse spatial audio signals and high-quality 3D human motion pairs. Bottom row: We develop a generative framework for human *MO*tion generation driven by *SP*atial *A*udio (MOSPA) to produce high-quality, responsive human motion driven by spatial audio. We note that the motion generation results are both realistic and responsive, effectively capturing both the spatial and semantic features of spatial audio inputs.

## Abstract

Enabling virtual humans to dynamically and realistically respond to diverse auditory stimuli remains a key challenge in character animation, demanding the integration of perceptual modeling and motion synthesis. Despite its significance, this task remains largely unexplored. Most previous works have primarily focused on mapping modalities like speech, audio, and music to generate human motion. As of yet, these models typically overlook the impact of spatial features encoded in spatial audio signals on human motion. To bridge this gap and enable high-quality modeling of human movements in response to spatial audio, we introduce the first comprehensive *S*patial *A*udio-Driven Human *M*otion (*SAM*) dataset, which contains diverse and high-quality spatial audio and motion data. For benchmarking, we develop a simple yet effective diffusion-based generative framework for human *MO*tion generation driven by *SP*atial *A*udio, termed *MOSPA*, which faithfully captures the relationship between body motion and spatial audio through an effective fusion mechanism. Once trained, MOSPA can generate diverse realistic human

---

*,† denote equal contributions and corresponding authors.

39th Conference on Neural Information Processing Systems (NeurIPS 2025).

motions conditioned on varying spatial audio inputs. We perform a thorough investigation of the proposed dataset and conduct extensive experiments for benchmarking, where our method achieves state-of-the-art performance on this task. Our code and model are publicly available at our *website*.

# 1   Introduction

Humans exhibit varying responses to different auditory inputs within a given space. For instance, when exposed to sharp, piercing sounds, individuals are likely to cover their ears and move away in the direction opposite to the sound source. Conversely, when the sound is soft and soothing, they may approach it out of curiosity or to investigate further. Therefore, generating realistic human motion for virtual characters to respond realistically to a variety of sounds in their environment is both a highly sought-after feature and is crucial for applications such as virtual reality, human-computer interaction, robotics, etc.

Unfortunately, while previous studies have extensively explored motion generation from action label [85, 29], text [93, 100, 77], music [78, 91, 46, 1], and speech [2, 97, 3, 102], human motion generation driven by *spatial audio* remains unexplored to the best of our knowledge. Unlike pure audio signals, e.g., music [70, 50, 78], speech [3, 97], the spatial audio signals not only does it encode semantics, but it also captures *spatial characteristics* that significantly influence body movements, requiring a specialized framework to accurately model motion responses to spatial audio stimuli.

To address this overlooked aspect, we propose to model the complex interactions between spatial audio inputs and human motion using a generative model. Since there is no such dataset tailored for this task, we first introduce the *SAM dataset* (Spatial Audio Motion dataset), which captures diverse human responses to various spatial audio conditions. This dataset is meticulously curated to include a wide range of spatial audio scenarios, enabling the study of motion conditioned on sound field variations. The *SAM dataset* has a total of more than 9 hours of motion, covering 27 common spatial audio scenarios and more than 70 audio clips. To ensure the diversity of the spatial audio, around 480 seconds of motion were captured for each audio clip at different positions in the character space. To ensure diverse motion responses to spatial audio, we introduce 20 distinct motion types (excluding motion genres) and 49 in total when including motion genres. We visualize samples from SAM in the top row of Fig. 1. See Appendix A for detailed statistics.

We further conduct benchmarking experiments on the proposed dataset, revealing the limitations of existing methods in this setting. To enable spatial audio-driven human motion generation, we introduce MOSPA, a simple yet effective framework tailored for this task. In real-world scenarios, human responses to sound are inherently influenced by spatial perception, intensity variations, directional cues, temporal dynamics, etc. Motivated by this, we generate motion by incorporating features extracted from the input spatial audio signals using [61]. Specifically, to capture intrinsic features across both temporal and spatial dimensions, we mainly utilize Mel-Frequency Cepstral Coefficients (MFCCs)[17] and Tempograms[28] to model the temporal characteristics of the audio. Additionally, we characterize the spatial audio by analyzing the root mean square (RMS) [61] energy, which quantifies signal intensity in audio processing.

These features enhance the effective modeling of the spatial and intensity variations of the spatial audio. To capture the distribution of spatial audio features and human motion dynamics effectively, we employ a diffusion-based generative model that ensures strong alignment between the two modalities—human motion and spatial audio signals. Leveraging diffusion models, MOSPA excels at modeling the complex interplay between spatial audio features and human motion. Besides, a residual feature fusion mechanism is employed to model the subtle influences of spatial audio on human movement.

Extensive evaluations on the SAM demonstrate that MOSPA achieves state-of-the-art performance on this task, outperforming existing baselines in generating realistic and diverse motion responses to spatial audio. Our contributions are summarized as follows:

- We introduce a novel task of spatial audio-conditioned motion generation and present the first comprehensive dataset SAM with over 9 hours of motion across diverse scenarios.
- We conduct extensive benchmarking and propose MOSPA, a diffusion-based generative framework tailored for modeling and generating diverse human motions from spatial audio.

- We achieve the SOTA performance on motion generation conditioned on spatial audio. Our dataset, code, and models will be publicly released for further research.

## 2    Related Work

**Spatial Audio.** Many studies have explored spatial audio modeling [98, 24, 99, 74, 37, 87, 44, 45]. For instance, [98] utilizes the natural synchronization between visual and audio modalities to learn models that jointly parse sounds and images without manual annotations. [24] leverages unlabeled audiovisual data to localize objects, such as moving vehicles, using only stereo sound at inference time. [99] reason about spatial sounds with large language models. Recently, spatial audio generation has been explored from text [74] and video [42]. [87] propose a method to model 3D spatial audio from body motion and speech. [37] presents a framework for spatial audio generation, capable of rendering 3D soundfields generated by human actions, including speech, footsteps, and hand-body interactions. Despite progress on spatial-audio tasks, generating human motion from spatial audio remains largely underexplored.

**Conditional Motion Generation.** Extensive efforts have been made into motion synthesis conditioning on user control signals [36, 72, 11, 79, 92], text [93, 100, 77, 30, 51, 53, 14], action [85, 29, 10, 39], music [78, 91, 46, 1], speech [2, 68, 38, 97, 3, 102], past trajectories [23, 9, 4, 90, 73], etc. We refer readers to [103] for a detailed survey of motion generation.

*Text-to-Motion.* Text-to-motion generation has recently gained popularity as an intuitive and user-friendly approach to synthesizing diverse human body motions. Generative pre-trained transformer frameworks have been utilized for text-to-motion generation [93, 57]. Subsequently, various generation techniques have been explored, including diffusion models [77], latent diffusion models [12], autoregressive diffusion model [69, 11], denoising diffusion GANs [100], consistency models [16], and generative masked modeling [31]. Recent advancements include the integration of motion generation with large language models [41, 94] and investigations into the scaling laws for motion generation [58, 22]. Recently, controllable text-to-motion generation has gained attention, enabling motion synthesis conditioned on both text prompts and control signals, e.g., target control points [84, 79].

*Music-to-Motion.* Recent advancements in Music-to-Motion generation have been made [21, 70, 48, 78, 27, 88, 71]. DanceFormer [47] adopts a two-stage approach, generating key poses for beat synchronization followed by parametric motion curves for smooth, rhythm-aligned movements. Bailando [70] utilizes a VQ-VAE to encode motion features via a choreographic memory module. [1] introduces a diffusion-based probabilistic model for motion generation, using a Conformer-based architecture. EDGE [78] also applies a diffusion model for dance generation and editing. Furthermore, multimodal approaches incorporating language and music enhance generation quality [27, 88, 15].

*Speech-to-Motion.* We mainly review studies on audio-driven motion (gesture) generation [97, 13, 3, 2, 102]. Early works are mostly based on GAN models [26, 54, 65, 89], while the recent attempts are mainly based on the generative diffusion model [102]. For instance, [97] proposes a generative retrieval framework leveraging a large language model to efficiently retrieve semantically appropriate gesture candidates from a motion library in response to input speech. [2] introduces a co-speech gesture synthesis method by employing a segmentation pipeline for temporal alignment and disentangling speech-motion embeddings to capture both semantics and subtle variations.

While audio signals have been widely used in music- and speech-to-motion tasks, human motion synthesis driven by spatial audio remains largely unexplored. As a result, data-driven methods are highly constrained by limited paired data. The goal of this paper is to develop a comprehensive dataset and a novel approach for high-quality spatial audio-driven motion synthesis.

## 3    SAM Dataset

We first introduce the Spatial Audio-driven Motion (SAM) dataset designed for human motion synthesis conditioned on spatial audio. We focus on binaural audio, a common form of spatial audio that aligns with human and (most) animal perception and can be readily applied to robotic platforms. SAM consists of more than 9 hours of human motions with corresponding binaural audio, and more than $4M$ frames, covering 27 common spatial audio scenarios and 20 common reaction types in daily life without counting the motion genres. The majority of the audio clips are sourced from the AudioSet [25], while only a small portion is extracted from publicly available YouTube videos

Table 1: Statistics of the SAM dataset. The SAM dataset encompasses 27 common daily spatial audio scenarios, over 20 reaction types excluding the motion genres, and 49 reaction types (see details in Appendix A). The number of subjects covered in SAM is 12, where 5 of them are female and the remaining 7 are male. It is also the first dataset to incorporate *spatial* audio information, annotated with Sound Source Location (SSL). The total duration of the dataset exceeds 34K seconds.

| Dataset | SSL | 3DJoint$_{pos/rot}$ | Model | Joints | Subjects | Seconds |
|---|---|---|---|---|---|---|
| Dance with Melody [75] | × | ✓/× | - | 21 | - | 5640 |
| DanceNet [104] | × | ✓/× | - | 55 | 2 | 3472 |
| AIST++ [48] | × | ✓/✓ | COCO/SMPL | 17/24 | 30 | 18694 |
| PopDanceSet [60] | × | ✓/✓ | COCO/SMPL | 17/24 | 132 | 12819 |
| FineDance [49] | × | ✓/✓ | SMPL & hand joints | 52 | 27 | 52560 |
| SAM (Ours) | ✓ | ✓/✓ | SMPL-X | 55 | 12 | 34356 |

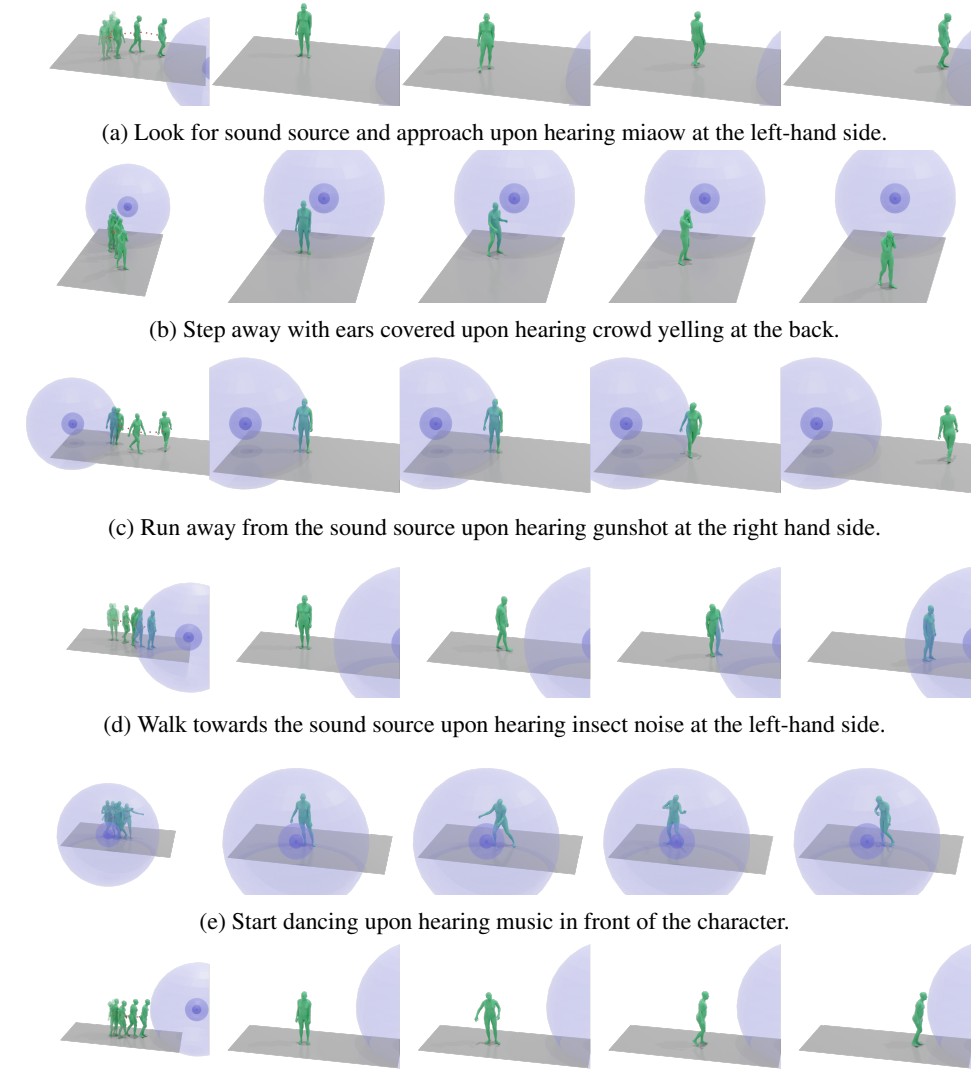

(a) Look for sound source and approach upon hearing miaow at the left-hand side.

(b) Step away with ears covered upon hearing crowd yelling at the back.

(c) Run away from the sound source upon hearing gunshot at the right hand side.

(d) Walk towards the sound source upon hearing insect noise at the left-hand side.

(e) Start dancing upon hearing music in front of the character.

(f) Look for sound source upon hearing the phone ring at the left-hand side.

Figure 2: Visualization of samples from SAM with expected motions annotated. Red dots indicate the actor's trajectory, while the blue sphere represents the sound source. The SAM dataset ensures high diversity by encompassing a broad spectrum of audio types and varying sound source locations.

or through manual recording. More detailed information can be found in Tab. 1 and Appendix A. Visualization results are in Fig. 2.

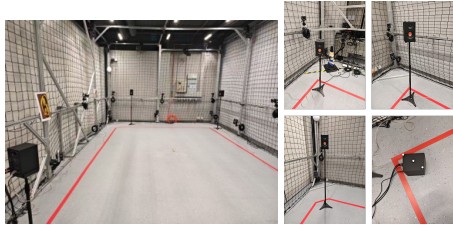
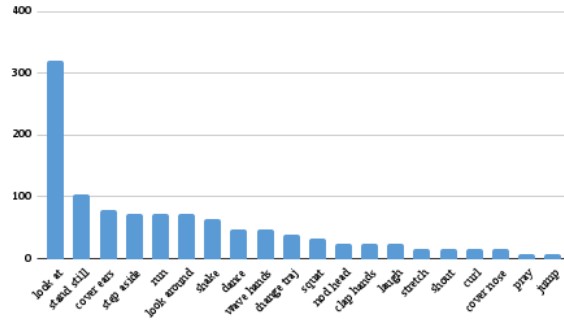

(i) Mocap environment.    (ii) The speakers.

Figure 3: Spatial audio-driven human motion data collection setup.

Figure 4: Statistics of action duration in the dataset.

**Data Capture Settings.** We utilize a Vicon motion capture system [56] to collect motion data and spatial audio signals. The motion capture is performed in a semi-open cage having a space of approximately $5m \times 10m \times 3m$, a structure covered by rope nets, with 28 mocap cameras mounted on the ropes and vertical supports recording at a frame rate of 120 Hz; See Fig. 3. The surrounding walls are standard painted concrete, resulting in a setting that resembles a typical indoor environment. In SAM, each audio clip is associated with 16 randomly sampled relative sound source locations, defined by combinations of different speakers and spatial positions relative to the subject. For each location, we capture three motion sequences corresponding to different reaction intensities: dull, neutral, and sensitive, resulting in a total of 48 motion sequences, each lasting 10 seconds. Fig. 4 shows the statistics of the approximate duration of the actions. The three motion genres define the varying degrees of responsiveness, decreasing from sensitive to dull. For instance, upon hearing an explosion, a dull individual might remain largely unreactive, whereas a sensitive one may immediately flee from the sound source. The total number of action types is 49. The percentage of action types covered within the dull, neutral, and sensitive motion genres are 28.57%, 34.69%, and 36.73%, respectively. To capture the binaural sound heard at the position of the actor, we employ 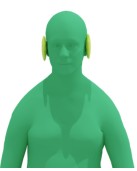 two microphones to record the audio at the ear positions of the actors separately; See the inset. The two microphones are connected to a Deity PR-2 recorder [18] that has been synchronized with the Vicon mocap system in advance using a timecode with a frame rate of 30 FPS. With this setting, the stereo sound at the position of the actor can be recorded and has an accurate alignment with the corresponding motion.

**Data Processing.** All motion and audio clips are precisely aligned. The motions are re-targeted and converted from the original BVH format in Vicon to the SMPL-X [63] format. SMPL-X is a parametric 3D human body model that encompasses the body, hands, and face, comprising $N = 10,475$ vertices and $K = 55$ joints. Given shape parameters $\beta$ and pose parameters $\theta$, the SMPL-X model generates the corresponding body shape and pose through forward dynamics. We extract the locations of the sound sources in each motion clip. The sound source locations are then transformed into the local space of the character aligned with the SMPL-X local coordinate system (a.k.a the local frame).

## 4   Method

We introduce MOSPA, a diffusion-based probabilistic model that serves as a baseline for this novel task of spatial audio-driven human motion generation. First, we extract spatial audio features $\mathbf{a}$ using a feature extractor [61]. During motion generation, the extracted spatial audio feature $\mathbf{a}$ is combined with the sound source location $\mathbf{s}$ and the motion genre $g$ as conditioning inputs. These inputs are passed to a denoiser $\mathcal{G}$, which is trained to reconstruct the original clean motion vector $\hat{\mathbf{x}_0}$ by denoising the given noisy motion vector $\mathbf{x_t}$ at time step $t$. Mathematically, we have $\hat{\mathbf{x}_0} = \mathcal{G}(\mathbf{x_t}, t; \mathbf{a}, \mathbf{s}, g)$.

### 4.1   Feature Representation

The two key vectors are the audio feature vector $\mathbf{a}$ and the motion vector $\mathbf{p}$. We carefully designed the structure of the two vectors in MOSPA.

**Spatial Audio Feature Extraction.** We first extract a range of audio features that capture intensity, temporal dynamics, and spatial characteristics. Inspired by [70], our feature set primarily includes *Mel-*

*frequency cepstral coefficients (MFCCs), MFCC delta, constant-Q chromagram, short-time Fourier transform (STFT) of the chromagram, onset strength, tempogram, and beats* [17, 67, 28, 61, 7, 20]. On top of these audio features, we additionally add the root mean square (RMS) energy $E_{rms}$ of the audio [61], and the active frames $F_{active}$ defined as $F_{active} = E_{rms} > 0.01$ to capture the distance information of the audio. The dimension of the audio feature vector for each ear is 1136. By concatenating the features from both ears, we obtain a combined feature vector **a** of dimension 2272. The detailed construction of the audio vector can be viewed in Appendix B.1.

**Motion Representation.** In this paper, we focus on body motion and leave the modeling of detailed finger movements to future work. Therefore, we exclude all the finger joints and retain only the first $J = 25$ body joints of the SMPL-X model [63]. In addition to the essential translation and joint rotations required for human pose representation, we introduce the residual feature fusion mechanism [30] to incorporate the global joint positions and the velocity of the joints to capture the nuanced difference in audio and further improve the accuracy of the generated samples. Each motion vector **x** is thus composed of the global positions $\mathbf{p} \in \mathbb{R}^{T \times (J \times 3)}$, the local rotations $\mathbf{r} \in \mathbb{R}^{T \times (J \times 6)}$ and the velocities $\mathbf{v} \in \mathbb{R}^{T \times (J \times 3)}$ of the joints (including the root), where $T = 240$ represents the number of frames in each motion sequence. The joint rotations are represented in the 6d format [101] to guarantee the continuity of the change ($\mathbf{x_0} = (\mathbf{p_0}, \mathbf{r_0}, \mathbf{v_0}), \mathbf{x} \in \mathbb{R}^{T \times (J \times 12)}$). The dimension of each motion vector is therefore 300.

## 4.2 Framework

Following [77, 11], the diffusion is modeled as a Markov chain process which progressively adds noise to clean motion vectors $\mathbf{x_0}$ in $t$ time steps, i.e.

$$q(\mathbf{x_t}|\mathbf{x_{t-1}}) = \mathcal{N}(\sqrt{\alpha_t}\mathbf{x_{t-1}}, (1-\alpha_t)I) \quad (1)$$

where $\alpha_t \in (0, 1)$. The model then learns to gradually denoise a noisy motion vector $\mathbf{x_t}$ in $t$ time steps, i.e. $p(\mathbf{x_{t-1}}|\mathbf{x_t})$. We directly predict the clean sample $\hat{\mathbf{x_0}}$ in each diffusion step $\hat{\mathbf{x_0}} = \mathcal{G}(\mathbf{x_t}, t; \mathbf{a}, \mathbf{s}, g)$, where $\mathbf{a}$ is the audio features, $\mathbf{s}$ is the sound source location and $g$ is the motion genre. This strategy, employed by [77, 11, 66, 84], has been proved to be more efficient and accurate than predicting the noise $\epsilon_t$, suggested by [35]. We employ an encoder-only Transformer to reverse the diffusion process and predict the clean samples. The timestep, motion, and conditioning signals are each projected into the same latent dimension using separate feed-forward networks. Random masks are applied to the audio features **a** and the sound source

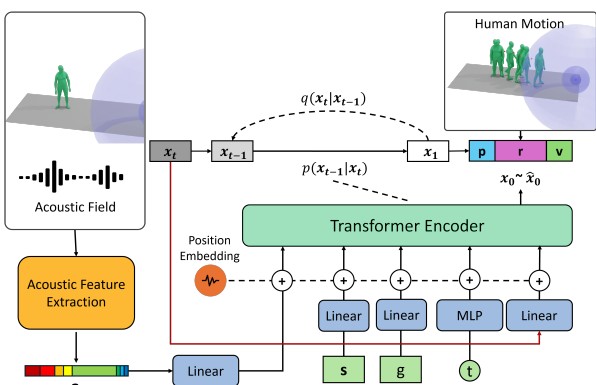

Figure 5: The framework of MOSPA. We perform diffusion-based motion generation given spatial audio inputs. Specifically, Gaussian noise is added to the clean motion sample $\mathbf{x_0}$, generating a noisy motion vector $\mathbf{x_t}$, modeled as $q(\mathbf{x_t}|\mathbf{x_{t-1}})$. An encoder transformer then predicts the clean motion from the noisy motion $\mathbf{x_t}$, guided by extracted audio features $\mathbf{a}$, sound source location (SSL) $\mathbf{s}$, motion genre $g$, and timestep $t$.

location (SSL) **s**, after which all components are concatenated to form the complete token sequence **z**. The tokens are positionally embedded afterward and input into a transformer to get the output $\hat{\mathbf{z}}$. The predicted clean sample is thus extracted from the last $T$ tokens of $\hat{\mathbf{z}}$ by inputting it to another feed-forward network, where $T = 240$ is the length of the motion; see Fig. 5.

## 4.3 Loss Functions

We train MOSPA using the following loss functions. A simple mean squared error (MSE) loss is applied to the original clean sample and the predicted clean sample as the main objective: $\mathbb{E}\|\hat{\mathbf{x_0}} - \mathbf{x_0}\|_2^2$. To guarantee the smooth variation on the predicted clean sample across frames, we also apply MSE loss to the rate of change of the vectors across frames: $\mathbb{E}\|\delta\hat{\mathbf{x_0}} - \delta\mathbf{x_0}\|_2^2$. Combining the two simple losses we have $\mathcal{L}_{data} = \mathbb{E}\|\hat{\mathbf{x_0}} - \mathbf{x_0}\|_2^2 + \mathbb{E}\|\delta\hat{\mathbf{x_0}} - \delta\mathbf{x_0}\|_2^2$. Geometric losses, encompassing position loss and velocity loss, are also incorporated, as we rely solely on joint

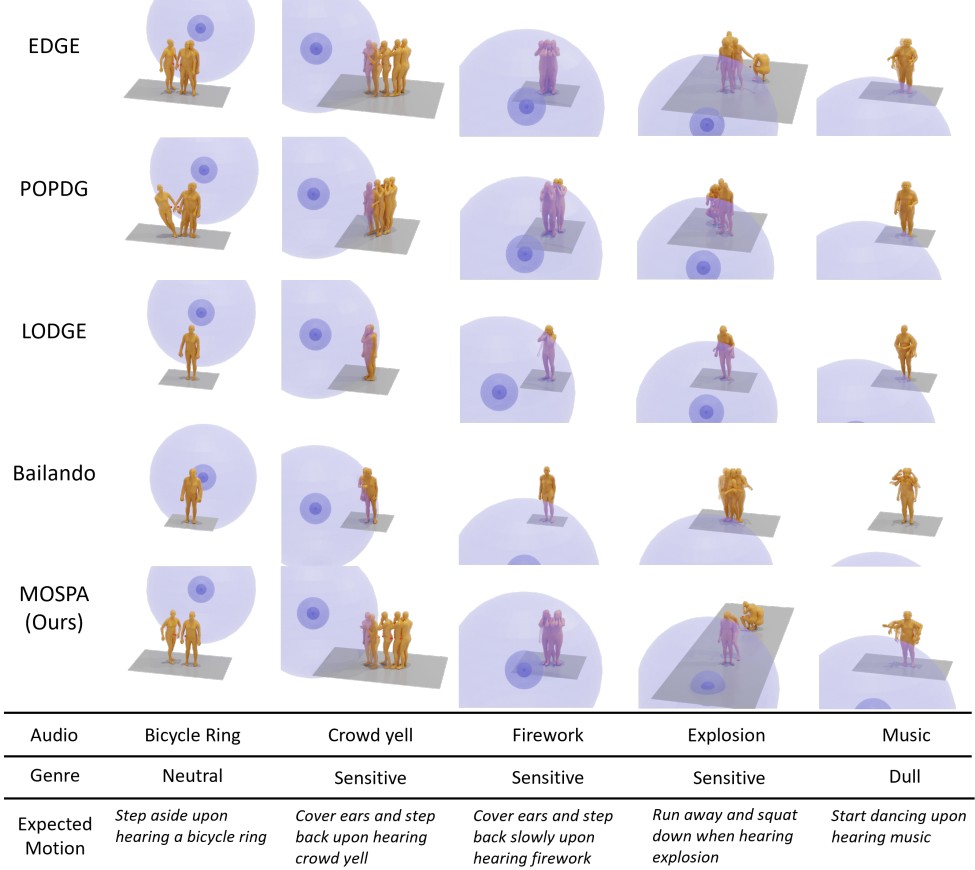

| Audio | Bicycle Ring | Crowd yell | Firework | Explosion | Music |
|---|---|---|---|---|---|
| Genre | Neutral | Sensitive | Sensitive | Sensitive | Dull |
| Expected Motion | *Step aside upon hearing a bicycle ring* | *Cover ears and step back upon hearing crowd yell* | *Cover ears and step back slowly upon hearing firework* | *Run away and squat down when hearing explosion* | *Start dancing upon hearing music* |

Figure 6: Qualitative comparison of state-of-the-art methods for the spatial audio-to-motion task. We visualize motion results from five cases. MOSPA produces high-quality movements that closely correspond to the input spatial audio. We provide Expected Motion as a description for reference.

rotations and translations in motion vectors to represent poses: $\mathcal{L}_{geo} = \mathbb{E}\|FK(\hat{\mathbf{x}_0}) - FK(\mathbf{x_0})\|_2^2 + \mathbb{E}\|\delta FK(\hat{\mathbf{x}_0}) - \delta FK(\mathbf{x_0})\|_2^2$. Furthermore, foot sliding is prevented by introducing the foot contact loss $\mathcal{L}_{foot}$ that measures the inconsistency in the velocities of the foot joints between the ground truth and the predicted motions.

We also incorporate trajectory loss and joint rotation loss to underscore their importance in achieving the training objectives and accelerate the convergence of the model, defined as $\mathcal{L}_{traj} = \mathbb{E}\|\hat{\mathbf{traj}}_0 - \mathbf{traj}_0\|_2^2 + \mathbb{E}\|\delta\hat{\mathbf{traj}}_0 - \delta\mathbf{traj}_0\|_2^2$ and $\mathcal{L}_{rot} = \mathbb{E}\|\hat{\mathbf{r}}_0 - \mathbf{r}_0\|_2^2 + \mathbb{E}\|\delta\hat{\mathbf{r}}_0 - \delta\mathbf{r}_0\|_2^2$ respectively, where $\mathbf{traj}$ is the trajectory vector of the motion sequence and $\mathbf{r}$ is the joint rotations represented in the 6d format [101]. Given that trajectory and joint rotations are inherently encoded within the motion vectors, these supplementary losses represent an overlap with the existing loss terms, effectively amplifying the emphasis on trajectory and joint rotation accuracy through increased weighting. Empirically, we observe that this implementation accelerates model convergence and facilitates correct displacement direction generation in motion sequences. In sum, the total loss is given by:

$$\mathcal{L} = \lambda_{data}\mathcal{L}_{data} + \lambda_{geo}\mathcal{L}_{geo} + \lambda_{foot}\mathcal{L}_{foot} + \lambda_{traj}\mathcal{L}_{traj} + \lambda_{rot}\mathcal{L}_{rot} \tag{2}$$

All loss weights ($\lambda$) are initialized set to 1. At epoch 5,000 of the total 6,000 training epochs, $\lambda_{traj}$ and $\lambda_{rot}$ are increased to 3, thereby intensifying the emphasis on trajectory and rotation accuracy.

## 4.4 Implementation Details

In MOSPA, the diffusion model is a transformer-based diffusion network [11, 77, 100]. The encoder transformer is configured with a latent dimension of 512, 8 heads, and 4 layers. We employ AdamW [55] as the optimizer with an initial value of $1 \times 10^{-4}$. The number of denoising steps used is 1000, and the noise schedule is cosine. The training phase concludes after $6,000$ epochs. Exceeding these recommended epoch counts may degrade model quality due to overfitting. The

Table 2: Quantitative evaluation on the SAM, where MOSPA achieves higher alignment with the GT motion while maintaining high diversity, as reflected by the metrics. The error bar is the 95% confidence interval assuming normal distribution, and → means the closer to Real Motion the better.

| Method | R-precision ↑ | | | FID ↓ | Diversity → | APD → |
|---|---|---|---|---|---|---|
| | Top1 | Top2 | Top3 | | | |
| Real Motion | $1.000^{\pm0.000}$ | $1.000^{\pm0.000}$ | $1.000^{\pm0.000}$ | 0.001 | $23.616^{\pm0.188}$ | 59.435 |
| EDGE [78] | $0.886^{\pm0.005}$ | $0.960^{\pm0.003}$ | $0.977^{\pm0.002}$ | 13.993 | $23.099^{\pm0.196}$ | 43.882 |
| POPDG [60] | $0.762^{\pm0.006}$ | $0.886^{\pm0.005}$ | $0.934^{\pm0.003}$ | 20.967 | $22.536^{\pm0.170}$ | 34.996 |
| LODGE [50] | $0.444^{\pm0.006}$ | $0.594^{\pm0.005}$ | $0.679^{\pm0.004}$ | 102.289 | $21.101^{\pm0.141}$ | 11.801 |
| Bailando [70] | $0.077^{\pm0.003}$ | $0.134^{\pm0.003}$ | $0.182^{\pm0.004}$ | 168.396 | $17.347^{\pm0.247}$ | 23.121 |
| MOSPA | $\mathbf{0.937^{\pm0.005}}$ | $\mathbf{0.984^{\pm0.002}}$ | $\mathbf{0.996^{\pm0.001}}$ | **7.981** | $\mathbf{23.575^{\pm0.188}}$ | **53.915** |

Table 3: Ablation study on MOSPA on the spatial audio-driven motion generation performance. The error bar is the 95% confidence interval assuming normal distribution, and → means the closer to real motions the better.

| Latent Dim | Head Num | Diff Steps | Genre | R-precision ↑ | | | FID ↓ | Diversity → | APD → |
|---|---|---|---|---|---|---|---|---|---|
| | | | | Top1 | Top2 | Top3 | | | |
| | Real Motion | | | $1.000^{\pm0.000}$ | $1.000^{\pm0.000}$ | $1.000^{\pm0.000}$ | 0.001 | $23.616^{\pm0.188}$ | 59.435 |
| 512 | 8 | 1000 | ✓ | $\mathbf{0.937^{\pm0.005}}$ | $0.984^{\pm0.002}$ | $0.996^{\pm0.001}$ | **7.981** | $\mathbf{23.575^{\pm0.188}}$ | 53.915 |
| 256 | 8 | 1000 | ✓ | $0.891^{\pm0.005}$ | $0.952^{\pm0.002}$ | $0.971^{\pm0.001}$ | 9.226 | $23.007^{\pm0.198}$ | 55.175 |
| 512 | 4 | 1000 | ✓ | $0.923^{\pm0.004}$ | $0.972^{\pm0.002}$ | $0.986^{\pm0.001}$ | 9.282 | $23.232^{\pm0.170}$ | **56.572** |
| 512 | 8 | 100 | ✓ | $0.930^{\pm0.004}$ | $0.980^{\pm0.002}$ | $0.991^{\pm0.001}$ | 8.456 | $23.351^{\pm0.177}$ | 49.824 |
| 512 | 8 | 4 | ✓ | $0.934^{\pm0.004}$ | $\mathbf{0.989^{\pm0.002}}$ | $\mathbf{0.998^{\pm0.001}}$ | 8.387 | $23.474^{\pm0.192}$ | 49.507 |
| 512 | 8 | 1000 | ✗ | $0.889^{\pm0.005}$ | $0.958^{\pm0.003}$ | $0.977^{\pm0.002}$ | 10.930 | $23.150^{\pm0.153}$ | 46.807 |

entire training process requires approximately 18 hours on a single RTX 4090 GPU with a batch size of 128.

# 5 Experiments

**Experiment Setup.** We use our SAM dataset to evaluate the spatial audio-driven motion generation task. As detailed in Sec. 3, it contains 9 hours of human motion with paired binaural audio and corresponding sound source locations, covering 27 common spatial audio scenarios and 20 common reaction types. The dataset is split into training, validation, and test sub-datasets at a common ratio of 8:1:1. Consequently, the training sub-dataset comprises 2,400 motion sequences, while the validation and test sub-datasets each contain approximately 300 motion sequences. To keep fair setting [10], the motions and the audio clips are both downsampled to the frame rate of 30 FPS. The character is rotated to face the negative y-axis and initially translated to the origin in the world space in all motion sequences, and the sound source locations (SSL) are transformed to the local space of the character in every single frame.

**Baselines and Metrics.** Our system is the first work to receive spatial audio as input to generate human motion results. To our best knowledge, as there is no other system achieving this, we made adaptations on other audio2motion methods, such as EDGE [78], POPDG [60], LODGE [50] and Bailando [70] by replacing their original audio input with our spatial audio feature as input. We evaluated four metrics, focusing on motion quality and diversity: **1)** *R-precision, FID, Diversity* These three metrics are calculated using the same setup proposed by [30]. Two bidirectional GRU are trained with a hidden size of 1024 for 1,500 epochs with a batch size of 64 to extract the audio features and the corresponding motion features, as suggested by [30]. Detailed implementation details of the feature extractor are provided in Appendix B.2. **2)** *APD* [19, 33] is calculated by

$APD(M) = \frac{1}{N(N-1)} \sum_{i=1}^{N} \sum_{\substack{j=1 \\ j \neq i}}^{N} \left( \sum_{t=1}^{L} \|\mathbf{s}_t^i - \mathbf{s}_t^j\|^2 \right)^{\frac{1}{2}}$, where $M = \{\hat{\mathbf{x}_i}\}$ is the set of generated

motion sequences, $N$ is the number of motion sequences in the set $M$, $L$ is the number of frames of each motion sequence, and $\mathbf{s}_t^i \in \hat{\mathbf{x}_i}$ is a state in the motion sequence $\hat{\mathbf{x}_i}$.

## 5.1 Comparisons

**Qualitative Results.** We demonstrate the qualitative comparison in Fig. 6. For the same input spatial audio, our methods show the superiority of producing high-quality and realistic response motion. Other methods often exhibit various limitations due to their unique model characteristics. EDGE [78] and POPDG [60] demonstrate relatively strong performance among the four baselines, sharing a diffusion-based foundation with MOSPA, despite differences in their encoding and decoding mechanisms. Their shortcomings in generated samples can primarily be attributed to model size and their strong focus on music-like audio. The bad performance of LODGE [50] is likely due to its specialization in long-term music-like audio, resulting in deficiencies when handling short-term audio information with abrupt feature changes. Similarly, Bailando [70] faces challenges in processing rapidly changing spatial audio. More critically, due to its separate training process for upper and lower body parts, Bailando occasionally produces distorted or disjointed motions when encountering sudden changes in spatial audio. Please watch our supplementary video for more results. Furthermore, we test MOSPA on out-of-distribution audio-source configurations. As shown in Fig. 8, it maintains motion quality and intent alignment, demonstrating robustness to unseen spatial setups.

**Quantitative Results.** The quantitative results are reported in Tab. 2. MOSPA achieves the best performance as shown by the lowest FID value and the highest R-precision values. Also, our generated motions exhibit the closest diversity and APD [19] values compared with the Real Motion, demonstrating the effectively balanced variation and precision. Bailando [70] has the worst performance among the four baselines in practice, as illustrated by the extremely high FID. The model possibly lacks the ability to perceive commonly heard sounds other than music and also the spatial information of the audio. Our method, overall speaking, still demonstrates competitive performance in spatial audio conditioned motion generation, which is proved by the low values in precision-related metrics and the high values in diversity-related metrics.

**User Study.** We conducted a user study with 25 participants to assess the perceptual quality of motion generation. Participants evaluated five models (MOSPA, EDGE, POPDG, LODGE, Bailando) alongside ground truth (GT), selecting the best motion for: **1)** *Human Intent Alignment*: Does the motion align with real-world intent? **2)** *Motion Quality*: Which has the highest movement quality? **3)** *GT Similarity*: Which best matches the GT motion? We provided GT motion and a textual description for reference. As shown in Fig. 7, MOSPA outperforms all baselines across all criteria, while LODGE and Bailando received the fewest selections, indicating limitations in generating realistic, semantically meaningful motions. See more details in Appendix C.

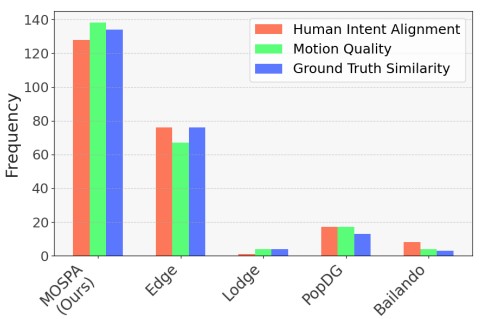

Figure 7: User study results. MOSPA outperforms other methods in intent alignment, motion quality, and similarity to ground truth. The bar chart shows the vote distribution across methods.

## 5.2 Ablation Study

We conducted ablation studies on the latent dimension, the number of attention heads, the diffusion step number, and the masking of motion genre, with results summarized in Tab. 3. All ablation experiments maintained consistent training epoch counts throughout.

**Latent Dimension.** The default latent dimension of MOSPA's encoder transformer is 512. In our study, reducing it to 256 slightly increases the APD [19] value but also degrades the R-precision and the FID, leading to an overall decline in model performance, as seen in row 1 and row 2 in Tab. 3.

**Number of Attention Heads.** We reduced the number of attention heads in MOSPA 's encoder transformer from 8 to 4, observing degradation in almost all of the metrics except a slight improvement in APD [19]. This reduction compromises overall model performance without yielding significant improvements in training efficiency, as seen in row 1 and row 3 in Tab. 3.

**Number of Diffusion Steps.** We evaluated MOSPA with varying diffusion step numbers, reducing it from 1000 to 100 and further to 4, as detailed in rows 1, 4, and 5 of Tab. 3. Fewer steps slightly degrade the performance as shown by the increase in FID and degradation in diversity, thereby lowering the upper limit of the power of the model.

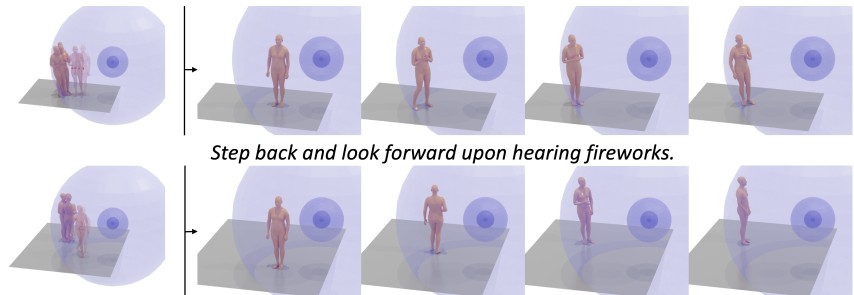

*Step back and look forward upon hearing fireworks.*

*Walk back and look around upon hearing vehicle horn.*

Figure 8: Test of MOSPA on out-of-distribution spatial audios. Descriptions of motions are provided for reference.

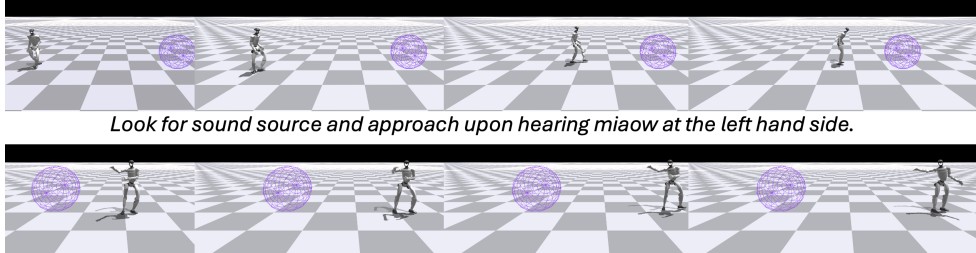

*Look for sound source and approach upon hearing miaow at the left hand side.*

*Get away from the sound source and wave hands when hearing noisy insects.*

Figure 9: Spatial audio-driven physically simulated humanoid robot control based on [34]. Descriptions of expected motion are provided for reference.

**Genre Masking.** Masking motion genres leads to a degradation in model performance across all metrics, as demonstrated in row 1 and 6 of Tab. 3. Motion genres are required to provide a guidance for the model on the intensity of the expected motions.

We evaluate the contribution of the extracted audio features by conducting an ablation study on the effectiveness of MFCC [17] and tempogram [28] features. As shown in Tab. 4, improvements in FID and R-precision—two key metrics for assessing generative quality and correspondence—demonstrate their significance in model.

Table 4: Ablation study on the effect of MFCC [17] and tempogram [28] features.

| MFCC | Tempogram | R-precision ↑ | | | FID ↓ |
|---|---|---|---|---|---|
| | | Top-1 | Top-2 | Top-3 | |
| ✓ | ✓ | $\mathbf{0.937}^{\pm 0.005}$ | $\mathbf{0.984}^{\pm 0.002}$ | $\mathbf{0.996}^{\pm 0.001}$ | **7.981** |
| ✗ | ✓ | $0.907^{\pm 0.004}$ | $0.967^{\pm 0.002}$ | $0.983^{\pm 0.002}$ | 9.070 |
| ✓ | ✗ | $0.917^{\pm 0.004}$ | $0.982^{\pm 0.002}$ | $0.994^{\pm 0.001}$ | 10.786 |

# 6 Conclusion

This introduces a novel task for enabling virtual humans to respond realistically to spatial auditory stimuli. We present a comprehensive SAM dataset, capturing human movement in response to spatial audio, and propose MOSPA, a diffusion-based generative model with an attention-based fusion mechanism. Once trained, MOSPA synthesizes diverse, high-quality motions that adapt to varying spatial audio inputs with binaural recording. Extensive evaluations show MOSPA achieves state-of-the-art performance on this task. *Limitations and Future Works.* Physical Correctness: While MOSPA generates diverse and semantically plausible motions, it lacks physical constraints, which may lead to physically implausible artifacts. Integrating physics-based control methods [19, 59, 76, 40, 95, 96, 62, 82] could improve motion realism and embodiment fidelity (see Fig. 9 for spatial audio-driven humanoid robot control). Body Modeling: This work focuses on body motion and omits finer-grained components such as hand gestures and facial expressions supported by SMPL-X [63]. Extending the model to full-body motion generation [57, 52, 86, 64, 5]—including hand motions—remains an important direction for future research. Scene Awareness: The current framework does not incorporate awareness of surrounding environments or physical scene geometry, limiting its ability to produce scene-consistent or contact-aware motions. Future extensions could integrate scene representations or affordance prediction [14, 80, 83, 8, 81] with spatial audio signals to enhance human motion generation.

## Acknowledgements

This work is partly supported by the Innovation and Technology Commission of the HKSAR Government under the ITSP-Platform grant (Ref: ITS/335/23FP) and the InnoHK initiative (TransGP project). Part of the research was conducted in the JC STEM Lab of Robotics for Soft Materials, funded by The Hong Kong Jockey Club Charities Trust. We are grateful to Yiduo Hao, Chuan Guo, Chen Wang, Zitong Lan, and Peter Qingxuan Wu for their insightful discussions and constructive feedback, which have been helpful to the development of this research.

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
