# OpenReview forum: "🎧MOSPA: Human Motion Generation Driven by Spatial Audio"
_NeurIPS.cc/2025/Conference — NeurIPS 2025 spotlight_

### Official Review · Reviewer_Jmh2 · 2025-06-29

**Clarity:** 3
**Significance:** 3
**Originality:** 4
**Rating:** 5
**Confidence:** 3

**Summary:**

The paper presents a human motion dataset comprised of spatial audio data, named the SAM dataset. The authors benchmark SAM using a diffusion-based, spatial audio-conditioned human motion generation framework (MOSPA) and demonstrate that MOSPA can be trained to generate human motions conditioned on spatial audio inputs, using the SAM dataset. The evaluation results show that MOSPA achieves good performance on the privacy-constrained HMR task.

**Questions:**

**Details of the SAM data.**

As SAM is comprised of recorded audio, the binaural cues may vary depending on the recording environment, which can be linked to the models’ performance. One might have questions about the following details:

1. What are the coordinates (locations) and the directions of the loudspeaker's location in the room, and what is the rationale for placing it that way?

2. Are the speakers/microphones omnidirectional? If not, what are their specifications?

3. How does the equipment used in the recording differ from the dummy head microphone, e.g. [N4], that is commonly used for binaural recordings? Is this equipment common for ‘binaural’ recordings?

4. What is shape of the room? How is the acoustics of the room (reverberant/anechoic?)

**Minor Comments.**

1. In L121, what does “16 randomly sampled relative locations” mean? Presumably that there were a total of 16 loudspeaker locations that were the source of the sound, from which the experiment randomly picked one to play the sound, and the audio clips in the SAM are paired with relative locations between the (stationary) loudspeakers from the (moving) human location?

    - MOSPA estimates $\mathbf{x}$ which consists of the global position ($\mathbf{p}$). Would there be any reason for encoding the sound source position as a relative position? (Perhaps due to the data sparsity in the sound sources’ global positions?)

1. Would it be possible to extract a person's position and the head orientation from the motion estimated by MOSPA?

    - If so, and if the original (mono) source sound is present, it can be rendered into binaural using efficient renderers such as [N5, N6]. It would be interesting to hear how well the binaurally rendered MOSPA’s sound source (say, $\tilde{\mathbf{a}}$) matches with the input binaural audio ($\mathbf{a}$). So that: $\mathbf{a}\stackrel{\mathcal{G}}{\longrightarrow}\hat{\mathbf{x}}_0 \stackrel{\text{[N5]}}{\longrightarrow} \tilde{\mathbf{a}}$ and then compare the similarity between the two binaural audios, e.g., $d(\tilde{\mathbf{a}}, \mathbf{a})$.

****
[N4] Gardner, B., & Martin, K. (1994, May). *HRFT Measurements of a KEMAR Dummy-head Microphone*.

[N5] Lee, J. W., & Lee, K. (2023, June). Neural fourier shift for binaural speech rendering. In *ICASSP 2023-2023 IEEE International Conference on Acoustics, Speech and Signal Processing (ICASSP)* (pp. 1-5). IEEE.

[N6] Roman, I. R., Ick, C., Ding, S., Roman, A. S., McFee, B., & Bello, J. P. (2024, April). Spatial scaper: a library to simulate and augment soundscapes for sound event localization and detection in realistic rooms. In *ICASSP 2024-2024 IEEE International Conference on Acoustics, Speech and Signal Processing (ICASSP)* (pp. 1221-1225). IEEE.

**Ethical Concerns:**

["NO or VERY MINOR ethics concerns only"]

**Final Justification:**

The authors resolved all of my concerns, and I maintain a score of 5.

**Limitations:**

yes

**Paper Formatting Concerns:**

No formatting concerns

**Quality:**

3

**Strengths And Weaknesses:**

**Strengths.**
1. To the best of the reviewer's knowledge, this paper is the first dataset for modeling the interaction between spatial information in audio and human motion.

2. The authors experimentally showed that motion synthesis models can be trained effectively on the proposed dataset, and by reporting a variety of metrics, they established a benchmark for the proposed task.

**Weaknesses.**

While the task of synthesizing human motion from audio is being discussed, the audio-related details have not been sufficiently elaborated.

- On the SAM dataset

    1. Please elaborate on the audio data processing configurations (e.g., sampling rate, STFT window/hop size, etc.)

    2. From the picture of the measurement environment in Figure 3, it seems that the sound was played through (stationary) speakers in the room and recorded from a microphone placed in the subject's ear. As the SAM data is comprised of the mic measurements, it seems to need a more specific description of the recording environment.

        - Please find more details in the Question section.

- On the MOSPA model

    1. What would be the rationale behind choosing an audio feature extraction as shown in Table B9? There is no mention of the selection criteria for the selected audio features, and no mention of features that are commonly used in the audio domain.

        - The task that MOSPA tackles seems to be closely related to the Sound Event Localization and Detection (SELD) problem, and many SELD studies primarily use log-mel spectrograms (for per-frequency energy) and generalized cross-correlations (for phase information) as audio features [N1, N2, N3].

        - ITD (Interaural Time Difference) and the ILD (Interaural Level Difference) are the most representative features for binaural audio cues, which are not mentioned in the text.

    2. It is unclear, at this moment, what the variable $\mathbf{s}$ means by reading the text “$\mathbf{s}$ is the sound source” in L181. However, assuming that it is meant to imply the (relative) location of the sound source, the question remains why this input is required.

        - Isn't it already possible to predict the source location from binaural recordings?

        - Doesn't needing to know the exact (relative) position of the sound source to synthesize motion limit its applicability?

        - Regarding the results in Table 3, the authors mention that entering location information improves FID and APD performance despite lowering other performance, so are these the two most important metrics?

****

[N1] Adavanne, S., Politis, A., Nikunen, J., & Virtanen, T. (2018). Sound event localization and detection of overlapping sources using convolutional recurrent neural networks. *IEEE Journal of Selected Topics in Signal Processing*, *13*(1), 34-48.

[N2] Wilkins, J., Fuentes, M., Bondi, L., Ghaffarzadegan, S., Abavisani, A., & Bello, J. P. TWO VS. FOUR-CHANNEL SOUND EVENT LOCALIZATION AND DETECTION.

[N3] Shimada, K., Simon, C., Shibuya, T., Takahashi, S., & Mitsufuji, Y. (2024). SAVGBench: Benchmarking Spatially Aligned Audio-Video Generation. *arXiv preprint arXiv:2412.13462*.

---

> ### Author Rebuttal · Authors · 2025-07-31
>
> We sincerely thank you for your insightful feedback and for recognizing both our proposed SAM dataset and the accompanying benchmark experiments. Please find our detailed responses to your questions and concerns below.
>
> ### **Audio data processing configurations**
> Thank you for your careful reading. We used a sampling rate of 30,720 Hz, an FFT window length of 2048, and a hop length of 256. The Short-Time Fourier Transform (STFT) windowing function is the Hann window [1]. We will add more details in the revision.
>
>
> ### **Audio feature extraction and explicit input of exact sound source location**
> We sincerely appreciate your detailed feedback. Our primary focus is spatial audio-conditioned motion synthesis, for which we leverage audio representations commonly used in music-to-dance pipelines [3, 4, 5], including MFCCs and tempograms. We agree with you that SELD-related features such as log-mel spectrograms, ITD/ILD, and cross-correlations are promising alternatives [N1–N3]; we are currently exploring these options and will include results in future updates.
>
> The explicit Sound Source Location (SSL) input was introduced to enhance the character animation system by providing directional cues for motion trajectory synthesis. Incorporating SSL is both practical and well-justified in virtual animation and simulation environments, where such spatial information is typically accessible and reliable. To address concerns about potential over-reliance on SSL, we conducted rigorous evaluations on out-of-distribution (OOD) audio samples (see Fig. 8 and Sec. 5.1). The results demonstrate that our model retains strong motion generation capabilities even when tested on audio inputs outside the SAM dataset. Additionally, we are currently conducting further experiments to generate motion directly from binaural inputs without explicit SSL supervision, and we will include these results in the revised version.
>
> For motion evaluation, people typically prioritize FID and R-precision to assess quality and audio-motion alignment, while Diversity and APD metrics quantify distributional consistency with ground truth. As stated in the paper: “R-precision, FID, Diversity, Multimodality. These four metrics are calculated using the same setup proposed by [2]. Two bidirectional GRU are trained with a latent size of 1024 for 1000 epochs to extract the audio features and the corresponding motion features.”
> We will expand the feature extractor architecture description in our revision.
>
> ### **Capture configurations**
>
> Thank you for your comments. To ensure data diversity in terms of distance and orientation, four speakers were positioned at the corners of the mocap area, each facing toward the center to minimize reverberation. As noted in the paper, motion capture sessions were conducted in an acoustically controlled empty room (5 m width × 10 m length × 3 m height). For each session, actors began from 16 randomly sampled initial positions and orientations. Because spatialized audio at the actor's location is essential, we placed microphones near the actors’ ears instead of using dummy head setups. In future work, we plan to further expand the dataset in both scale and diversity.
>
> ### **References**
> [1] RB Blackman and JW Tukey. The measurement of power spectra dover publications. Inc, New York, 1958.
> [2] Chuan Guo, Shihao Zou, Xinxin Zuo, Sen Wang, Wei Ji, Xingyu Li, and Li Cheng. Generating diverse and natural 3d human motions from text. In Proceedings of the IEEE/CVF conference on computer vision and pattern recognition, pages 5152–5161, 2022.
> [3] Ruilong Li, Shan Yang, David A Ross, and Angjoo Kanazawa. Ai choreographer: Music conditioned 3d dance generation with aist++. In Proceedings of the IEEE/CVF international conference on computer vision, pages 13401–13412, 2021.
> [4] Li Siyao, Weijiang Yu, Tianpei Gu, Chunze Lin, Quan Wang, Chen Qian, Chen Change Loy, and Ziwei Liu. Bailando: 3d dance generation by actor-critic gpt with choreographic memory. In Proceedings of the IEEE/CVF Conference on Computer Vision and Pattern Recognition, pages 11050–11059, 2022.
> [5] Wenlin Zhuang, Congyi Wang, Jinxiang Chai, Yangang Wang, Ming Shao, and Siyu Xia. Music2dance: Dancenet for music-driven dance generation. ACM Transactions on Multimedia Computing, Communications, and Applications (TOMM), 18(2):1–21, 2022.

---

> > ### Comment · Reviewer_Jmh2 · 2025-08-02
> >
> > Thanks for the response, and I'm glad it helped inform improvements that will be reflected in future revisions.
> >
> > While most of my concerns have been resolved, a concern remains about the acoustic details of the capture environment. This is because this paper discusses *spatial audio-driven* motion capture, and at this moment, it's unclear how well the spatial audio was recorded in the well-established environment.
> >
> > ****
> >
> > To add a follow-up question on the authors' response:
> >
> > > To ensure data diversity in terms of distance and orientation, four speakers were positioned at the corners of the mocap area, each facing toward the center to minimize reverberation.
> >
> > I understand that placing the speakers close to the corners can have acoustic advantages, not necessarily minimizing reverbs, but minimizing some interferences such as Speaker Boundary Interference Response (SBIR). The reverb is more dominantly affected by the walls' material properties (usually characterized by acoustic reflection impedances), which was not mentioned in the paper.
> >
> > > As noted in the paper, motion capture sessions were conducted in an acoustically controlled empty room (5 m width × 10 m length × 3 m height).
> >
> > It remains unclear what kind of acoustic treatment was used in the room, and the text doesn't mention anything beyond its dimensions. What does "controlled" mean? Was there a specific treatment used to eliminate room reverberation? Should the readers interpret it as a general reflective wall, which implies that this recording is a spatial audio recording of a typical room acoustic?
> >
> > ****
> >
> > Regarding data diversity, I believe there is still room for improvement in terms of the spatial distribution of sources, and a more diverse speaker array would have made this a better resource. However, since improving this diversity is unlikely to be achieved during the discussion period, I believe that at least a detailed description of the acoustic environment is necessary. I maintain a score of 5.

---

> > > ### Author Response · Authors · 2025-08-02
> > >
> > > Thank you very much for your insightful comments and follow-up. To clarify the acoustic environment: the motion capture was conducted in a semi-open cage structure covered by rope nets, with mocap cameras mounted on the ropes and vertical supports. The surrounding walls are standard painted concrete, resulting in a setting that resembles a typical indoor environment rather than a professionally treated acoustic studio. As such, the room acoustics exhibit moderate reverberation, consistent with common office or residential spaces — a setting we believe better aligns with real-world deployment scenarios and helps minimize the domain gap between collected data and downstream usage. We are currently investigating the specific coating materials used on the walls (as well as the size of the entire room) and will include these details in the revised version. Yes, we also agree that reverberation is inherently present and often unavoidable in indoor environments. To mitigate its impact on Sound Event Localization and Detection, we incorporate more information like Sound Source Location information during training - this helps stabilize directional cues and reduces the influence of reverberation on the resulting motion generation.  In the revision, we will include a detailed description of the room structure, surface materials, mitigation strategies, etc.
> > >
> > > We fully agree that the current speaker configuration could be improved in terms of spatial diversity. We will provide a more detailed description of the acoustic environment in the revision, and we plan to explore more diverse speaker arrangements and improved room acoustics in future data collection.
> > >
> > > Once again, thank you for your thoughtful feedback (along with that of the other reviewers) has meaningfully contributed to improving this research work.

---

### Official Review · Reviewer_njb5 · 2025-06-29

**Clarity:** 3
**Significance:** 3
**Originality:** 3
**Rating:** 5
**Confidence:** 4

**Summary:**

This paper proposes a new dataset, SAM, and a novel model, MOSPA, for human motion generation using spatial sound. The proposed dataset was recorded in an indoor setting and includes sound source locations, motion genres, binaural audio, and corresponding human motion data. The proposed method extracts acoustic features and, together with genre and source location information, utilizes a diffusion model to generate motion that corresponds to the audio signals. Experimental results on the SAM dataset demonstrate that the proposed approach produces more realistic and diverse motions compared to existing music-based baselines.

**Questions:**

(Major) Regarding the first weakness on robustness to recording environments: even if motion capture data collection is infeasible in diverse settings, qualitative results in environments with significantly different layouts, reverberation characteristics, or outdoor conditions could considerably impact the perceived effectiveness of the method and potentially alter the overall rating.

(Minor) A more detailed discussion on which features or preprocessing steps led to the performance differences between the proposed method and EDGE would greatly strengthen the paper’s technical contribution.

(Minor) While the paper focuses on spatial audio as the primary input, it would be valuable to discuss the potential for incorporating language information to enable more controllable motion generation.

**Ethical Concerns:**

["NO or VERY MINOR ethics concerns only"]

**Final Justification:**

The authors appropriately demonstrated the importance of acoustic features by responding to my comments with supporting experimental results. While it is true that there are some issues, such as limited diversity in recording environments and missing language captions, I believe the dataset—combining spatial audio and human motion—offers a high degree of novelty and contributes meaningfully to the advancement of the field. For this reason, I have raised my score to Accept.

**Limitations:**

yes

**Quality:**

3

**Strengths And Weaknesses:**

Strengths:
- Compared to prior work on gesture generation aligned with speech or music, the proposed task of generating motion based on spatial audio offers a high degree of novelty.
- The paper is well-written and very easy to follow.
- The proposed dataset, SAM, provides valuable insights into how humans react to spatial audio and is likely to contribute meaningfully to the advancement of the field.
- The experimental results demonstrate that the proposed method outperforms existing approaches both quantitatively and qualitatively, highlighting the effectiveness of the proposed acoustic features and the additional conditioning information such as genre.

Weaknesses:
- The data collection environment is limited to a single indoor setting. It is well known that directional acoustic features are highly dependent on room layout and reverberation characteristics [1, 2]. Since the proposed dataset is recorded in only one room, it remains unclear whether the proposed method would generalize well to different indoor environments or outdoor scenarios. This limitation is particularly relevant for scenarios such as “insect” or “firework,” where the input audio would naturally be expected to originate from outdoor environments.
- The proposed model does not take language as input, which limits its controllability. Although the effectiveness of genre-based control is demonstrated, there is no mechanism for more fine-grained control over the content or style of the generated motion.
- The paper lacks in-depth analysis regarding the performance gap between models. While the proposed method is relatively simple in structure (which is not a weakness in itself), the discussion around its performance compared to other diffusion-based methods such as EDGE, is insufficient. The authors only mention:
> "Their shortcomings in generated samples can primarily be attributed to model size and their strong focus on music-like audio."

However, it would be desirable to include a more concrete analysis, such as which features in the proposed method were particularly effective for non-music sounds.

[1] Oumi, Yusuke, et al. "Acoustic-based 3D Human Pose Estimation Robust to Human Position." BMVC2204.

[2] Wang, Mason, et al. "Soundcam: A dataset for finding humans using room acoustics." NeurIPS2023.

---

> ### Author Rebuttal · Authors · 2025-07-31
>
> We sincerely appreciate the time and expertise dedicated to evaluating our work. We are pleased to see your recognition of the SAM dataset's novelty and the methodological rigor demonstrated in our experiments and paper. Next, we provide our comprehensive responses below:
>
> ### **Robustness to recording environments**
> Thank you for your insightful feedback. We fully acknowledge the importance of evaluating the method’s robustness under diverse recording environments, such as varying room layouts, reverberation characteristics, or outdoor settings, which can influence acoustic perception as well as motion generation.
>
> Currently, we focus on indoor environments to ensure controlled and consistent data quality. We carefully designed the data collection process to minimize confounding factors. For example, during recording, speakers were positioned directly in front of the actors to reduce acoustic ambiguity and mitigate potential reverberation effects. As reported in Section 5.1, our model demonstrates strong performance on both in-distribution and out-of-distribution (OOD) samples within these indoor settings.
>
> That said, we agree that broader environmental variation is a valuable next step. In future work, we plan to extend our data collection and evaluation to include diverse spatial environments—including outdoor scenes and complex indoor layouts.
>
> ### **In-depth analysis of the performance gap between methods**
>
> We agree that a more detailed analysis would be valuable and will include it in our revision. For example, our spatio-temporal audio conditioning leverages both the sound’s location and temporal structure, which is particularly effective for handling non-music audio events such as sudden or ambient sounds.
>
> ### **Which features in the proposed method were particularly effective for non-music sounds**
> Thank you for these pertinent suggestions. We conducted ablation studies on audio feature selection by training MOSPA under two modified conditions:
> 1. Exclusion of MFCC features.
> 2. Exclusion of tempogram features.
>
> Quantitative results demonstrate that omitting either feature set significantly degrades motion synthesis quality and accuracy relative to ground truth. Specifically, tempogram has a larger contribution to the generation of precise motion sequences. These ablation studies confirm their essential role in our audio-to-motion translation framework.
> We will include the aforementioned experiments and discussion in the revision once available.
>
> | MFCC | Tempogram | FID↓ | R-1↑ | R-2↑ | R-3↑ |
> |------|-----------|------|------|------|------|
> | ✔    | ✔        | $7.981$ | $0.937^{±0.005}$ | $0.984^{±0.002}$ | $0.996^{±0.001}$ |
> | ✘    | ✔        | $9.070$ | $0.907^{±0.004}$ | $0.967^{±0.002}$ | $0.983^{±0.002}$ |
> | ✔    | ✘        | $10.786$ | $0.917^{±0.004}$ | $0.982^{±0.002}$ | $0.994^{±0.001}$ |
>
> ### **Potential for incorporating language information**
> Thanks for this insightful suggestion. Yes! Integrating language control with spatial audio input represents a highly promising multimodal research direction. While our current focus is on spatial audio-driven motion synthesis, we plan to explore multimodal integration in future work.

---

> > ### Comment · Reviewer_njb5 · 2025-08-03
> >
> > **Robustness to recording environments**
> >
> > Unfortunately, I could not understand your answer clearly.
> > > For example, during recording, speakers were positioned directly in front of the actors to reduce acoustic ambiguity and mitigate potential reverberation effects.
> >
> > I checked the supplementary material, and it seems that in many of the videos, the sound source is placed behind or to the side of the subject (e.g., "Step aside upon hearing bicycle ring from the back"). Are all of these considered OOD samples? Regarding the training data, was the sound source always placed in front of the subject? If so, I don’t see the necessity of using DoA features.
> >
> > **On Additional Ablation Studies**
> >
> > Thank you for showing us insightful results. I totally understand the importance of these acoustic features.

---

> > > ### Author Response · Authors · 2025-08-04
> > >
> > > Thank you very much for your kind follow-up.
> > >
> > > We apologize for the lack of clarity in our previous explanation regarding the recording setup. To clarify, the speakers were positioned in various directions around the actor—in front, behind, to the left, and to the right—but were consistently oriented to face the actor in order to reduce acoustic ambiguity. We appreciate your careful reading and will revise this phrasing to be clearer in the final version.
> > >
> > > Regarding the OOD evaluation: by out-of-distribution (OOD), we refer to novel combinations of both sound source locations and audio content. For instance, while the training set includes speaker positions at the front, back, left, and right of the actor, we test OOD spatial generalization by synthesizing interpolated positions, such as placing the sound source at the rear-diagonal direction, which was never explicitly seen during training. We also evaluate on entirely new audio samples not used in the training phase. These OOD conditions are designed to assess the model’s ability to generalize across both spatial and semantic variations.
> > >
> > > Thank you again for your valuable comments—they have helped us clarify important aspects of our setup and improve the clarity of our presentation. We will revise the paper accordingly.

---

### Official Review · Reviewer_wAmb · 2025-07-01

**Clarity:** 2
**Significance:** 2
**Originality:** 2
**Rating:** 5
**Confidence:** 4

**Summary:**

This paper focus on the task of human motion generation driven by spatial audio, which aims to explore the relationship between spatial audio cues and human motion. The paper presents the Spatial Audio Motion (SAM) dataset, which contains annotations of sound source locations and different reaction intensities. The authors introduce the MOSPA, a diffusion-based model that achieves SOTA performance in generating realistic and responsive human motion.

**Questions:**

1. Given that the SAM dataset contains a limited set of 12 subjects and 27 spatial audio scenes, how do you account for potential biases in motion responses?
2. Have you performed ablation studies to understand the respective contributions of MFCCs, temporal maps, and RMS energy to the final representation?
3. Have you tested MOSPA on other publicly available audio-to-motion benchmarks, even if they lack spatial annotations?
4. How does the model perform with spatial audio inputs that differ significantly from the training distribution in SAM?

**Ethical Concerns:**

["NO or VERY MINOR ethics concerns only"]

**Final Justification:**

The issues I raised have all been addressed, so I am happy to raise my score to 5. I apologize for the late reply.

**Limitations:**

yes

**Quality:**

2

**Strengths And Weaknesses:**

Strengths:
1. This paper aim to addresses the new task of spatial audio-driven human motion generation and proposes a framework to model the relationship between spatial audio cues and human motion.
2. It propose a new SAM dataset, which offers above 8 hours of annotated motion and spatial audio data, enabling research in this area. The proposed diffusion-based model effectively captures audio-motion interplay and shows strong performance.

Weaknesses:
1. It only involve 12 participants and 27 audio scenarios may restrict diversity and generalizability.
2. The model architecture appears relatively simple except the spatial audio feature extraction pipeline, and the absence of ablation studies makes it hard to assess the contributions of individual components.
3. The lack of evaluation on external datasets raises concerns about robustness and real-world applicability.
4. Its evaluation is limited to a single dataset, with limited sample size and scene coverage, and lacks ablation studies on the proposed diffusion model. There are concerns about its generalization and robustness due to the lack of testing outside of the SAM dataset. More extensive benchmarks and deeper analysis of the model would help strengthen this work.

---

> ### Author Rebuttal · Authors · 2025-07-31
>
> Thank you sincerely for your thorough and insightful comments. We are grateful to your recognition of the research direction, as well as the proposed framework and dataset. Regarding the weaknesses you pointed out and the questions you raised, we would like to provide the following explanations:
>
> ### **Audio feature extraction**
> Thank you for your valuable suggestions. Despite inherent differences between musical and non-musical audio characteristics, certain features remain effective for general sound representation, like MFCCs and tempogram. Following established music-to-dance methodologies [2, 3, 4], we employ common audio features including MFCCs and tempograms. Building upon this foundation, we augment the feature set by concatenating binaural audio features and incorporating RMS energy as a distance indicator.
> Ablation studies confirm that models trained without MFCCs or tempograms exhibit significantly reduced motion accuracy compared to ground truth:
>
> | MFCC | Tempogram | FID↓ | R-1↑ | R-2↑ | R-3↑ |
> |------|-----------|------|------|------|------|
> | ✔    | ✔        | $7.981$ | $0.937^{±0.005}$ | $0.984^{±0.002}$ | $0.996^{±0.001}$ |
> | ✘    | ✔        | $9.070$ | $0.907^{±0.004}$ | $0.967^{±0.002}$ | $0.983^{±0.002}$ |
> | ✔    | ✘        | $10.786$ | $0.917^{±0.004}$ | $0.982^{±0.002}$ | $0.994^{±0.001}$ |
>
> ### **Lack of evaluation on external datasets raises concerns about robustness and real-world applicability?**
> To our knowledge, the SAM dataset is the *first* motion-capture-based audio-motion paired dataset with recorded spatial audio information, while uniquely encompassing both musical and non-musical audio. Existing datasets only focus on musical content and *lack spatial localization*, making them unsuitable for spatial audio-driven motion synthesis tasks.
>
> ### **Ablation studies on the proposed diffusion model and testing outside of the SAM dataset**
> Thank you for your insightful feedback. As shown in Tab. 3 of the paper, we have conducted extensive ablation studies on the configuration of the proposed diffusion model MOSPA. We have also tested the model on out-of-distribution samples which are not in the SAM dataset: *"Furthermore, we test MOSPA on out-of-distribution audio-source configurations. As shown in Fig. 8, it maintains motion quality and intent alignment, demonstrating robustness to unseen spatial setups."*. We will also include additional out-of-distribution (OOD) samples in our test set to better assess generalization.
>
> ### **Restrict diversity and generalizability of the dataset?**
> As shown in Appendix Table A4, we respectfully argue that our 27 audio scenarios cover a wide range of real-life situations—including noisy and quiet environments, sudden transient sounds, and steady background noise. To further ensure diversity and generalizability, all 12 subjects performed different motions based on both the type of sound and its spatial relation to the subject’s initial position and orientation.
>
> Compared to existing music-to-dance datasets, the SAM dataset exhibits substantially greater diversity and scale, as summarized in Table 1 of the main paper. With 12 subjects and 33,600 seconds of motion-audio pairs, SAM significantly surpasses many comparable datasets—for instance, Music2Dance [4] includes only 2 subjects, and AIST++ [2] provides 18,694 seconds of paired data. This volume supports a more comprehensive representation of varied motion behaviors.
>
> We will continuously expand the dataset in both diversity and scale to support broader research and application needs.
>
> ### **References**
> [1] Chuan Guo, Shihao Zou, Xinxin Zuo, Sen Wang, Wei Ji, Xingyu Li, and Li Cheng. Generating diverse and natural 3d human motions from text. In Proceedings of the IEEE/CVF conference on computer vision and pattern recognition, pages 5152–5161, 2022.
> [2] Ruilong Li, Shan Yang, David A Ross, and Angjoo Kanazawa. Ai choreographer: Music conditioned 3d dance generation with aist++. In Proceedings of the IEEE/CVF international conference on computer vision, pages 13401–13412, 2021.
> [3] Li Siyao, Weijiang Yu, Tianpei Gu, Chunze Lin, Quan Wang, Chen Qian, Chen Change Loy, and Ziwei Liu. Bailando: 3d dance generation by actor-critic gpt with choreographic memory. In Proceedings of the IEEE/CVF Conference on Computer Vision and Pattern Recognition, pages 11050–11059, 2022.
> [4] Wenlin Zhuang, Congyi Wang, Jinxiang Chai, Yangang Wang, Ming Shao, and Siyu Xia. Music2dance: Dancenet for music-driven dance generation. ACM Transactions on Multimedia Computing, Communications, and Applications (TOMM), 18(2):1–21, 2022.

---

> > ### Comment · Reviewer_wAmb · 2025-08-05
> >
> > The issues I raised have all been addressed, so I am happy to raise my score to 5. I apologize for the late reply.

---

> ### Comment · Area_Chair_mFtG · 2025-08-05
>
> Dear Reviewer wAmb,
>
> The deadline for author-reviewer discussion period is approaching. We kindly ask you to review the authors' rebuttal. Please provide your feedback soon. Thank you.
>
> Best,
>
> AC

---

### Official Review · Reviewer_peZR · 2025-07-02

**Clarity:** 4
**Significance:** 3
**Originality:** 3
**Rating:** 5
**Confidence:** 5

**Summary:**

This paper presents a new task-human motion generation conditioned on spatial audio, along with an 8-hour mocap dataset (SAM) and a spatial audio conditioned motion generation model (MOSPA). Extensive experiments demonstrate the proposed MOSPA can achieve better performance compared to the baseline methods. Qualitative results show responsive motion driven by spatial audio.

**Questions:**

1. In Abstract #line17, what is the “privacy-constrained HMR task”? I did not find any reference in the paper.
2. What is the motivation for including the sound source location as an explicit input condition? From the basic principles of spatial audio, one would expect a model to implicitly infer the source location from spatial audio features. Given that explicit sound source locations are typically unavailable in real-world scenarios, why is this input still deemed necessary? How do the authors interpret the effect of the sound source location on the quality of the generated motion? For example, does the character fail to orient correctly (e.g., look in the wrong direction) without this explicit input?

**Ethical Concerns:**

["NO or VERY MINOR ethics concerns only"]

**Final Justification:**

All of my concerns have been well addressed. I believe this work makes valuable contribution to the human motion and animation community, and should be presented in NeurIPS.

**Limitations:**

Yes

**Paper Formatting Concerns:**

There’s no formatting concern.

**Quality:**

3

**Strengths And Weaknesses:**

## Strengths
1. Human motion generation driven by spatial audio is an interesting and novel direction, with strong potential for real-world applications. A high-quality, mocap-based paired dataset is a crucial foundation for advancing this line of research. The proposed SAM dataset is collected using a professional motion capture system, with well-documented collection procedures and comprehensive dataset statistics. This dataset represents a significant and valuable contribution to the field.
2. The model design is very clear and achieve good performance, which set a good benchmark for future research.
3. The paper presents comprehensive experiments that effectively demonstrate the proposed method and design choices. The demos on out-of-distribution (OOD) spatial audio and simulated character control are also commendable and add practical value to the work.

## Weaknesses
1. The generated motion exhibits quite a few foot-sliding artifacts, which is worse than motion generation results in other tasks, e.g. music2dance, text2motion.
2. The methodological contribution appears relatively minor, as the core innovation lies in the spatial audio feature extraction module. It would be worthwhile to include ablation studies on the spatial audio features to better understand their impact and validate the effectiveness of different features.

---

> ### Author Rebuttal · Authors · 2025-07-31
>
> We sincerely appreciate your valuable feedback and your recognition of our work's contributions, including the introduction of a high-quality mocap-based paired dataset for spatial audio conditioned motion synthesis tasks, a clearly designed model which achieves good performance, and the comprehensive experiments we have done on the model and the dataset. We have carefully considered all the comments and weaknesses you have pointed out. Below, we address each point systematically:
>
>
> ### **Foot-sliding artifacts**
>
> We acknowledge that foot-sliding artifacts are a well-known issue across various (kinematics-based) motion synthesis frameworks, including both AR-based approaches [8, 4] and Diffusion-based models [6, 1, 10]. We fully agree that improving physical plausibility is a valuable and promising direction for future research, especially given its significance for perceptual realism—as highlighted in prior works [7, 9, 2].
>
> That said, our current work prioritizes addressing what we believe are more immediate challenges in spatial-audio-driven motion generation: namely, limited training data, the alignment between spatial audio and generated motion, and overall motion quality. While physical realism remains an important goal, we consider it part of a broader research trajectory beyond the current scope:) As outlined in our Future Work section, we intend to incorporate physics-informed constraints and contact-aware refinement strategies to further enhance the physical realism of generated motions.
>
> ### **Audio feature extraction**
> Thank you for your insightful comments. Although music-like audio and general non-musical audio exhibit various differences, certain common audio features—such as MFCCs and tempograms—still represent fundamental audio characteristics across domains. Consequently, inspired by prior music-to-dance research [5, 11, 3], we utilize many common audio features. Building upon these foundations, we augment the feature set to incorporate spatial information: specifically, we concatenate channel-specific features for the left and right ears to further emphasize spatial information and introduce RMS energy to implicitly represent relative audio distance. See detailed results below:
>
> | MFCC | Tempogram | FID↓ | R-1↑ | R-2↑ | R-3↑ |
> |------|-----------|------|------|------|------|
> | ✔    | ✔        | $7.981$ | $0.937^{±0.005}$ | $0.984^{±0.002}$ | $0.996^{±0.001}$ |
> | ✘    | ✔        | $9.070$ | $0.907^{±0.004}$ | $0.967^{±0.002}$ | $0.983^{±0.002}$ |
> | ✔    | ✘        | $10.786$ | $0.917^{±0.004}$ | $0.982^{±0.002}$ | $0.994^{±0.001}$ |
>
>
> ### **Q: What is the motivation for including the sound source location as an explicit input condition?**
> Thank you for your insightful comments. The explicit Sound Source Location (SSL) input was originally introduced to support the character animation system by providing informative guidance for motion trajectory synthesis, particularly under acoustically challenging conditions such as strong reverberation. Moreover, incorporating SSL is both practical and justified in virtual animation and simulation environments, where such information is typically accessible and reliable.
> Nevertheless, our dataset naturally accommodates scenarios without SSL input, enabling the model to function effectively under both SSL-present and SSL-absent conditions - a promising direction for embodied AI and robotics research.
>
> Regarding concerns about potential over-reliance on SSL: We conducted rigorous experiments on out-of-distribution (OOD) samples (see Fig. 8 and Sec. 5.1 of the paper). Results demonstrate that the model maintains reasonable motion generation capability using audio features outside the SAM dataset.
>
> Additional validation includes:
> - Training/evaluation with fully masked SSL, yielding motions generally well-aligned with ground truth
> - Directional loss implementation (measuring angular deviation between predicted/ground-truth motion vectors), which partially addresses orientation errors when the SSL is not feasible
>
> We will add more experiments and discussion to the revision.
>
> ### **Writing**
> We are sorry for the misunderstanding caused by "privacy-constrained". We will fix this in our revision.
>
> ### **References**
> [1] Xin Chen, Biao Jiang, Wen Liu, Zilong Huang, Bin Fu, Tao Chen, and Gang Yu. Executing your commands via motion diffusion in latent space. In Proceedings of the IEEE/CVF conference on computer vision and pattern recognition, pages 18000–18010, 2023.
> [2] Yifeng Jiang, Jungdam Won, Yuting Ye, and C Karen Liu. Drop: Dynamics responses from human motion prior and projective dynamics. In SIGGRAPH Asia 2023 Conference Papers, pages 1–11, 2023.
> [3] Ruilong Li, Shan Yang, David A Ross, and Angjoo Kanazawa. Ai choreographer: Music conditioned 3d dance generation with aist++. In Proceedings of the IEEE/CVF international conference on computer vision, pages 13401–13412, 2021.
> [4] Shunlin Lu, Jingbo Wang, Zeyu Lu, Ling-Hao Chen, Wenxun Dai, Junting Dong, Zhiyang Dou, Bo Dai, and Ruimao Zhang. Scamo: Exploring the scaling law in autoregressive motion generation model. In Proceedings of the Computer Vision and Pattern Recognition Conference, pages 27872–27882, 2025.
> [5] Li Siyao, Weijiang Yu, Tianpei Gu, Chunze Lin, Quan Wang, Chen Qian, Chen Change Loy, and Ziwei Liu. Bailando: 3d dance generation by actor-critic gpt with choreographic memory. In Proceedings of the IEEE/CVF Conference on Computer Vision and Pattern Recognition, pages 11050–11059, 2022.
> [6] Guy Tevet, Sigal Raab, Brian Gordon, Yonatan Shafir, Daniel Cohen-Or, and Amit H Bermano. Human motion diffusion model. arXiv preprint arXiv:2209.14916, 2022.
> [7] Ye Yuan, Jiaming Song, Umar Iqbal, Arash Vahdat, and Jan Kautz. Physdiff: Physics-guided human motion diffusion model. In Proceedings of the IEEE/CVF international conference on computer vision, pages 16010–16021, 2023.
> [8] Jianrong Zhang, Yangsong Zhang, Xiaodong Cun, Yong Zhang, Hongwei Zhao, Hongtao Lu, Xi Shen, and Ying Shan. Generating human motion from textual descriptions with discrete representations. In Proceedings of the IEEE/CVF conference on computer vision and pattern recognition, pages 14730–14740, 2023.
> [9] Yufei Zhang, Jeffrey O Kephart, Zijun Cui, and Qiang Ji. Physpt: Physics-aware pretrained transformer for estimating human dynamics from monocular videos. In Proceedings of the IEEE/CVF Conference on Computer Vision and Pattern Recognition, pages 2305–2317, 2024.
> [10] Wenyang Zhou, Zhiyang Dou, Zeyu Cao, Zhouyingcheng Liao, Jingbo Wang, Wenjia Wang, Yuan Liu, Taku Komura, Wenping Wang, and Lingjie Liu. Emdm: Efficient motion diffusion model for fast and high-quality motion generation. In European Conference on Computer Vision, pages 18–38. Springer, 2024.
> [11] Wenlin Zhuang, Congyi Wang, Jinxiang Chai, Yangang Wang, Ming Shao, and Siyu Xia. Music2dance: Dancenet for music-driven dance generation. ACM Transactions on Multimedia Computing, Communications, and Applications (TOMM), 18(2):1–21, 2022.

---

> > ### Comment · Reviewer_peZR · 2025-07-31
> >
> > Thanks for the response. All of my concerns have been well addressed. I believe this work makes valuable contribution to the human motion and animation community, and should be presented in NeurIPS. I will raise the score from 4 to 5.

---

> > > ### Author Response · Authors · 2025-08-01
> > >
> > > We sincerely thank you for your constructive comments and the time you invested in reviewing this work. As promised, we will further clarify the discussed points and improve the paper in the revision. Thank you again.

---

### Note · Authors · 2025-08-12

We sincerely thank the reviewers for their thoughtful and constructive feedback and the ACs for their support. We are encouraged by their recognition of the strengths and contributions of our work, and we deeply value the time and effort they invested in their evaluations. For clarity, we will refer to Reviewers peZR, wAmb, njb5, and Jmh2 as R1, R2, R3, and R4, respectively, throughout this response.

In particular, we are encouraged by the reviewers’ recognition of our contributions: addressing the new task of human motion synthesis conditioned on spatial audio (R1, R2, R3, R4); proposing the SAM dataset—a high-quality, mocap-based resource designed to model the relationship between spatial audio and human motion (R1, R2, R3, R4); introducing a clear and effective framework (R1, R2, R3, R4); and establishing a benchmark to support future research (R1, R4). We also appreciate the reviewers’ positive feedback on the presentation of our work, noting that we have conducted comprehensive experiments demonstrating the value of our approach (R1, R3) and that the paper is clear and easy to follow (R3).

During the rebuttal, we primarily addressed: the choices for audio feature extraction (R1, R2, R3); the motivation for including explicit Sound Source Location (SSL) as an input condition (R1, R4); the foot-sliding artifacts (R1); the robustness of the proposed model (R2); the generalizability of the SAM dataset (R2); the dataset’s robustness to different recording environments (R3); the provision of detailed audio data processing configurations (R4); and additional details regarding the capture environments (R4).

We are deeply grateful for the reviewers’ constructive comments, which have provided valuable guidance for improving our work. We will be sure to integrate these points into our revision, and we would like to express our sincere thanks once again for your time, effort, and thoughtful assessment.

---

### Decision · Program_Chairs · 2025-09-17

**Decision:**

Accept (spotlight)

**Comment:**

This paper introduces MOSPA, a diffusion-based framework for generating realistic human motions driven by spatial audio, addressing a gap in existing research that primarily focuses on non-spatial audio modalities like speech or music. The authors present the SAM dataset, the first comprehensive collection of spatial audio and corresponding human motions, featuring over 8 hours of data across diverse scenarios. MOSPA leverages spatial audio features such as MFCCs, tempograms, and RMS energy to model the relationship between audio signals and motion responses for high-quality synthesis. Comprehensive experiments validating MOSPA’s ability to generate diverse, realistic motions aligned with spatial audio inputs, including robustness tests on out-of-distribution samples.

This paper has several strengths. The task and dataset are pioneering, addressing an underexplored intersection of spatial audio perception and human motion synthesis. A new dataset SAM is introduced, which is meticulously collected with professional mocap systems, diverse audio scenarios, and clear documentation, filling a critical resource gap. Furthermore, MOSPA’s design is well-motivated, with strong empirical results demonstrating superior performance over baselines. However, the reviewer raised several concerns. The generated motions exhibit quite a few foot-sliding artifacts. The dataset is limited to 12 subjects and 27 audio scenarios in a single indoor environment, raising questions about generalizability to outdoor or acoustically varied settings. The diffusion framework itself is straightforward, lacking deeper architectural insights. The motivation for including explicit Sound Source Location (SSL) as an input condition needs clarification.

During the rebuttal phase, most concerns have been addressed by the authors. Reviewer peZR and Jmh2 questioned the over-reliance on SSL. The authors clarify that SSL aids trajectory synthesis, but OOD tests show robustness without SSL. Reviewer peZR questioned unrealistic foot motions. The authors acknowledged this as a known challenge, deferring physics-based refinements to future work. Reviewer wAmb raised concern about the dataset generalizability. The authors highlighted SAM’s scale vs. prior datasets, with plans to expand diversity.

After the rebuttal, all reviewers leaned toward acceptance. This paper stands out for its novelty and potential to catalyze research in audio-driven animation. With the introduction of a new task with real-world applications, it has the potential to inspire follow-up work in this field.